# Expert Opinions about Barriers and Facilitators to Physical Activity Participation in Ghanaian Adults with Type 2 Diabetes: A Qualitative Descriptive Study

**DOI:** 10.3390/sports11070123

**Published:** 2023-06-21

**Authors:** Mohammed Amin, Debra Kerr, Yacoba Atiase, Yusif Yakub, Andrea Driscoll

**Affiliations:** 1Centre for Quality and Patient Safety, Institute for Health Transformation, Faculty of Health, School of Nursing and Midwifery, Deakin University, 221 Burwood Highway, Burwood, VIC 3125, Australia; aminmo@deakin.edu.au (M.A.); d.kerr@deakin.edu.au (D.K.); 2National Diabetes Management and Research Centre, Korle-Bu Teaching Hospital, University of Ghana Medical School, Accra P.O. Box GP 4236, Ghana; yatiase@ug.edu.gh; 3Faculty of Medicine and Health, The University of Sydney, Science Rd., Camperdown, NSW 2050, Australia; yyak3902@uni.sydney.edu.au

**Keywords:** physical activity, exercise, type 2 diabetes, healthcare professionals, barriers, facilitators, Ghana, adults, qualitative study

## Abstract

Most adults with type 2 diabetes mellitus (T2DM) do not meet their physical activity (PA) goals despite its importance in improving their health outcomes. Our study aim was to explore the opinions of healthcare professionals regarding barriers and facilitators to PA participation in Ghanaian adults with T2DM. Using qualitative descriptive design, data were collected through semi-structured interviews with 13 healthcare professionals experienced in diabetes management in Ghana. Three main themes relating to PA barriers and facilitators were identified in a thematic analysis: health system-related factors, healthcare practitioner factors, and patient factors. Inadequate accessibility to physical therapists and therapy centres hindered the provision of PA programs. Nurses and doctors lacked sufficient knowledge and training on effective PA interventions for individuals with T2DM. Time constraints during patient consultations limited discussions on PA, while the cost associated with accessing physical therapy posed a significant challenge. Patients often disregarded PA advice from physical therapists due to their reliance on doctors, and some perceived PA as irrelevant for diabetes treatment. Despite these barriers, healthcare professionals expressed belief in PA facilitators, including integrating physical therapists and diabetes educators into diabetes care, providing structured exercise resources, improving curriculum planning to emphasise PA in health science education, and addressing knowledge gaps and misconceptions. Overall, this study highlights patient-related and healthcare system-related factors that influence PA behaviour in Ghanaian adults with T2DM. Findings from this study should inform the development of tailored PA programs for this population.

## 1. Introduction

Type 2 diabetes mellitus (T2DM) is a metabolic disorder characterised by high blood sugar resulting from poor insulin secretion, action or both [1]. Globally, T2DM is the most common type of diabetes and accounts for 90–95% of total diabetes mellitus cases [2]. There is strong evidence that poorly controlled T2DM results in long-term complications and comorbidities including cardiovascular diseases (CVD), blindness, end-stage renal impairment and lower limb amputations [3,4]. Cardiovascular disease accounts for about 80% of the total mortality in individuals with T2DM [5], contributing substantially to healthcare costs and quality of life [6]. People with T2DM have twice the chance of developing CVD later in life compared to people without diabetes [7].

African countries are experiencing a rise in non-communicable diseases including diabetes and this trend is partly blamed on the increasing physical inertia among the population [8]. Type 2 diabetes is a major health problem in Ghana [9]. Approximately 90% of Ghanaian adults with T2DM have multiple comorbidities which increases their risk of developing CVD [9]. Ghana, a low–middle-income African country with a population of 30.8 million [10], faces varying poverty rates of 37.9% in rural areas and 10.6% in urban areas [11]. Significant strides have been made in providing healthcare access to Ghanaians, with the implementation of the National Health Insurance Scheme (NHIS) in 2005 [12]. Approximately 40% of the population is actively enrolled in the scheme which covers approximately 95% of medical conditions and services [12]. Despite these efforts, the prevalence of diabetes in Ghana has been steadily increasing, with a current rate of 6.5% [13] compared to 0.2% in the 1950s [14]. Given that Ghana and other sub-Saharan African countries are projected to have the highest diabetes burden [15], the region has the smallest healthcare budget, i.e., barely 1% of the global health budget, to fight the disease [16]. The burden of infectious diseases bedeviling the region coupled with the rising prevalence of non-communicable diseases like T2DM puts pressure on the health budgets of the various countries [17]. Adopting a healthy lifestyle would contribute to preventing diabetes-related complications, thereby easing the pressure on health budgets [18].

Physical activity (PA) plays an important role in T2DM management, particularly in improving insulin resistance, mental health, and reducing free fatty acid levels [19], as well as improving blood pressure and promoting weight loss [20,21]. Additionally, there is strong evidence that PA improves the quality of life in adults with T2DM [22,23,24]. Despite these benefits, most people with T2DM, especially in low–middle-income countries do not meet their PA goals [25]. In Ghana, physical inertia accounts for 19% and 3% of CVDs and diabetes, respectively, and about 20% of non-communicable disease-related deaths [26,27]. Despite the high likelihood of Ghanaian adults with T2DM accepting a PA program [24], evidence shows that only 21.4% of them achieve their PA targets [28]. This indicates a gap between the willingness to participate in PA programs and the actual ability to meet the recommended levels of PA. Further research and interventions are needed to address the factors influencing PA participation in this population.

Considerable progress has been made in understanding the factors influencing PA participation among Ghanaians. In a study conducted in Ghana, Tuakli-Wosornu et al. [29] explored the perception of Ghanaian women on PA. Barriers to PA identified in that study included time constraints to exercise, work/family obligations, and limited access to exercise facilities [29]. Common facilitators identified were the benefit of weight loss, positive health outcomes, and increased energy [29]. Another study involving Ghanaians identified hypertension barriers including inadequate social support, lack of time to exercise, limited exercise facilities, and high costs associated with accessing those facilities [30]. However, facilitators of PA participation identified were perceived health benefits, physician/family support, and access to community or faith-based fitness groups [30]. Furthermore, a scoping review focusing on non-communicable diseases and PA in Ghana revealed significant PA barriers and facilitators in Ghanaian adults [31]. The review identified barriers including limited knowledge and access to facilities, absence of suitable exercise spaces, and insufficient knowledge of PA guidelines and implementation methods [31]. However, peer and family influence, group-based activities, and advice from HCPs positively impact PA among older adults [31].

Progress in PA research is limited by the unique socio-cultural, health, and environmental factors associated with each population. Increasing PA participation requires the identification of the unique barriers and facilitators relative to each population. In our previous study (currently in draft), we reported PA barriers and facilitators in Ghanaian adults with T2DM, from the perspective of consumers. However, collecting data from healthcare professionals (HCPs) who have strong working relationships with their patients may offer some perspectives that otherwise might not have been identified by consumers. Therefore, the aim of this study was to explore the opinions of HCPs regarding barriers and facilitators to PA participation among people with T2DM in Ghana.

## 2. Materials and Methods

### 2.1. Study Design

A qualitative descriptive design was employed to explore HCPs’ opinions on the barriers and facilitators to PA among people with T2DM in Ghana. The design aligns with the constructivist paradigm and was adopted to help provide a direct description of the different experiences and perceptions that HCPs have about patients’ PA behaviour.

The study was conducted in accordance with the principle of the Declaration of Helsinki [32]. This study was reviewed and approved by the Deakin University Human Research Ethics Committee (Reference: HEAG-H 237_2020) and the Korle-Bu Teaching Hospital Institutional Review Board (Reference: KBTH-IRB/000146/2020). Participants provided verbal informed consent. The Standards for Reporting Qualitative Research (SRQR) was adopted for reporting this study.

### 2.2. Study Setting

The study was conducted in the Greater Accra and Ashanti regions of Ghana. The study sites included two specialist diabetes clinics (five participants), two district hospitals (two participants), research institutions (three participants), and a physical therapy centre (three participants). Some participants had affiliations with both hospitals and research institutions. Specialist diabetes clinics and physical therapy centres are mostly available at the teaching and some regional hospitals. Those hospitals receive referrals from smaller district hospitals and polyclinics which mostly lack specialist capacity.

### 2.3. Recruitment

Purposive sampling was used and included 13 HCPs with professional experience in caring for people with diabetes in Ghana. Using snowball sampling, we continued recruitment until such a time that data saturation was achieved. Participants met the following inclusion criteria: (a) employed in Ghana as a nurse or medical doctor in a diabetes healthcare clinic; or (b) employed in Ghana as an exercise science researcher, exercise physiologist, physiotherapist, occupational therapist, sports scientist, and/or sports physician; and (c) at least 2 years working experience in Ghana, relative to their area of practice. The sample size for the study was based on theoretical data saturation.

### 2.4. Interview Schedule

The authors developed a semi-structured interview guide (Appendix A), which was informed by a literature review and aligned with the study objectives. The interview guide consisted of two sections: a demographic information section and a section containing questions pertaining to the PA behaviour of Ghanaian adults with type 2 diabetes. Participants were asked open-ended questions such as: What are the barriers to physical activity participation among people with type 2 diabetes? What can be done to support them to overcome those barriers? How best can people with type 2 diabetes be supported to reduce their perceived physical activity barriers/challenges? Probing questions were used to seek clarification and expansive responses.

Interviews were audio recorded via Zoom and later transcribed verbatim by one researcher. Data were collected between February and May 2021.

### 2.5. Data Analysis

Participant characteristics are summarised using descriptive statistics. Each transcript was assigned a code depending on the order in which the participant enrolled in the study. For example, the first physiotherapist, physician, nurse, and exercise physiologist were assigned PT1, P1, N1, and EP1, respectively. Data were analysed using NVivo software (version 12).

Data were analysed using thematic analysis as described by Braun and Clarke [33]. The following six steps were involved in the thematic analysis: (a) familiarization with data including reading and re-reading the transcripts; (b) identification of initial codes; (c) identifying preliminary themes; (d) developing a thematic map determined in a consensus meeting by all authors; (e) refinement of labels; and (f) selecting compelling extracts relative to the study objectives. The authors closely examined the transcripts to gain a comprehensive understanding of the data. Codes that exhibited similarities were then grouped into sub-themes. The researchers critically reviewed the sub-themes and organized them into overarching main themes. Through collaborative discussions, the authors achieved consensus on the definitions and labels attributed to each theme.

### 2.6. Rigour

To ensure trustworthiness, participants were selected based on predetermined inclusion criteria. Further, an appropriate sampling technique and sample size were ensured. Participants’ views were explored in a 30 to 60 min individual interview to gain a true reflection of their social reality. Privacy was maintained during those interviews to allow participants to freely express themselves. Further, the authors bracketed their potential biases during data collection and analysis. Moreover, a detailed description of the study context is provided in this paper to allow readers to judge whether the study findings were transferable to other contexts. An audit trail was maintained.

## 3. Results

A sample of 13 participants participated in this study. Characteristics of participants are outlined in Table 1.

Three main themes were identified: health system-related factors, healthcare practitioner factors, and patient factors as shown in Table 2.

### 3.1. Health System-Related Factors

Health system-related factors encompass those aspects of the Ghanaian healthcare system that can either hinder or support exercise participation for individuals with T2DM. Our study identified factors such as the limited time during patient consultations to promote PA, insufficient access to exercise therapists and programs, absence of clear-cut PA guidelines from the Ghana Health Service, and the financial barrier posed by the cost of physical therapy. Additionally, other significant factors involve highlighting the roles of physical therapists and diabetes educators in diabetes care, enhancing the curriculum to incorporate PA as an essential component of nursing and medical education, ensuring the availability of structured exercise resources, and addressing knowledge gaps and misconceptions surrounding PA.

Lack of time to assess, treat and manage clients was viewed as a barrier to the provision of comprehensive care, including the promotion of PA. Healthcare professionals lamented the large patient numbers which limits the amount of time to discuss PA with patients during consultation.


*With our system, we are burdened with work. … There are constraints on the amount of time you can give one patient. There are 100 people waiting at your consulting room.*
—P3

Some participants expressed disappointment that clinical exercise physiologists are not employed in health facilities to support medical rehabilitation for diabetes patients. In their view, there is overdependence on nurses, physicians and physiotherapists for medical rehabilitation, which may not meet the needs of people with T2DM.


*In our healthcare facilities, there is a unit for physiotherapy but there is no unit for medical rehabilitation.... I appreciate the work of physiotherapy and doctors but when it has to do with diabetes, it is more [important for] exercise scientists, exercise physiologists.*
—EP1

Some participants expressed concern about the inadequate physical therapy centres in the Ghanaian healthcare system. They also expressed worry about the lack of PA programs for people with T2DM.


*[There are] very few places you could send patients to… If you have a patient that you want to refer for specific physical therapy, for diabetes, you are not even sure where to refer them to.*
—P1

Most HCPs raised concerns about the lack of clear-cut PA guidelines specifically relevant to patients with diabetes, as prescribed by the Ghana Health Service. Nurses and physicians disclosed that they mostly deliver PA advice based on opinion rather than scientific guidelines.


*You see, GHS (Ghana Health Service) is not up and doing……unlike advanced countries where there’s standard and comprehensive diabetes care guidelines, over here [in Ghana] our guidelines lack details. Particularly on exercise, there’s no clarity…..each doctor does his/her own thing.*
—P3

The cost of exercise therapy was identified as a potential barrier for some individuals. Exercise therapy is not currently covered by the National Health Insurance Scheme in Ghana. Given the low economic status of most patients, HCPs believe that their patients cannot afford exercise therapy.


*If a patient must meet the exercise therapist, that patient should be ready to pay. The low economic status of our people will not even permit them to [pay].*
—PT2

Participants elucidated several factors that they believed may contribute to the promotion of PA participation. Most HCPs expressed optimism about the potential benefits for patients in meeting their PA guidelines by emphasising the roles of physical therapists and diabetes educators in the delivery of diabetes care in Ghana. They believe patients will be better supported by these specialists in meeting their PA goals.


*Mostly general nurses give patient education at the diabetes clinic. Sometimes, at the district level, this education is done by Healthcare Assistants. But I think these roles should be handled by certified Diabetes Educators or the physio [physiotherapy] team.*
—N2

Participants advocated for improvement in the curriculum of nursing and medical schools to incorporate comprehensive content on PA. They expressed the belief that the education and training they receive in school do not equip them with the necessary knowledge and skills to deliver competent PA advice to patients. This deficiency in education creates a disadvantage for patients who do not receive the needed support from HCPs.


*The training of the healthcare providers should give much consideration to physical activity and exercise. For example, looking at the curriculum I think that they should have credit hours for physical therapy or exercise therapy….and those subjects should be handled by experts in exercise science.*
—EP3

Participants suggested that access to structured exercise resources, such as exercise brochures or videos relevant to diabetes patients, will facilitate their role in delivering PA advice to patients.


*I think we can get the [exercise] experts to develop something like a leaflet on exercise, with illustrations on how to perform those exercises……. it’ll guide us on what to do and it makes our work easier.*
—N3

### 3.2. Healthcare Practitioner Factors

Healthcare practitioner factors refer to the characteristics and behaviours of healthcare professionals directly engaged in the care of patients with T2DM, which can either impede or facilitate PA participation among Ghanaians with T2DM. Our study identified factors such as a lack of evidence-based knowledge to educate patients on PA, over-emphasising medication and diet in diabetes management, and provision of continuous professional development for nurses and doctors.

Most participants expressed the view that some HCPs lack knowledge about the type of PA for improving health outcomes for individuals with T2DM. This lack of evidence-based knowledge leads to inadequate patient education regarding the type and benefits of PA.


*Most of our health professionals do not know much about exercise. Most don’t know the difference between aerobic exercise and anaerobic exercise, strength training … a lot of doctors will advise their [diabetes] patients to avoid strength training and only do aerobic exercise when the research shows that it is best to do both.*
—P2

Healthcare professionals acknowledged that there is a notable focus on specific aspects of the diabetes regimen, namely, medication, while paying little or no attention to PA. Consequently, patients may not receive comprehensive education, support, and guidance regarding PA. This imbalance in emphasis could lead to a gap in knowledge and understanding of the importance of PA, as well as a lack of strategies and resources to incorporate regular physical activity into diabetes management.


*When they [patients] are coming to the hospital, they are coming because their medications have finished….and most of the time, the doctors focus on managing diabetes through pharmacological method. They don’t want to think about diet, they don’t want to think about the exercise therapist.*
—PT2

When HCPs were asked about the support needed to enhance their competence and confidence in the delivery of PA advice and programs, they suggested continuous professional development programs aimed at enhancing their knowledge and skills in implementing PA programs.


*I think we should upgrade ourselves in terms of PA in diabetes management. We could be supported with in-house training, or workshops specific to chronic diseases and modifiable lifestyle behaviours.*
—P2

### 3.3. Patient Factors

Patient factors are characteristics related to patients with T2DM that either promote or hinder their PA participation. In this study, several patient-related factors were identified, including patients’ tendency to overly rely on advice solely from physicians, resulting in the disregard of advice from other HCPs including physical therapists. Furthermore, there is a prevailing perception among some patients that PA is not a viable treatment option for diabetes. Addressing knowledge gaps and misconceptions surrounding PA also emerged as an important patient-related factor.

HCPs reported that some patients seek herbal treatment and spiritual healing in contrast to recommended treatment at the hospital. These individuals do not consider PA as a treatment option.


*Some believe that with diabetes you must go with herbal treatment. …. Some people go to see soothsayers [fortune-tellers], some go about collecting concoctions [alternate treatment].*
—PT1

Our finding sheds light on a specific behaviour within the Ghanaian context, where patients place significant trust and reliance on the expertise of physicians, potentially overlooking the valuable insights and recommendations offered by other members of the healthcare team. By disregarding advice from other HCPs, patients may not receive comprehensive care that addresses all aspects of their condition.


*Our patients largely depend on the doctor, the doctor and the doctor. The input of other healthcare professionals is minimised. So it is just the doctor at the forefront. And the doctor is limited; he prescribes medication and advises the patient. But with regards to other aspects of management, the doctor might not be able to help.*
—EP1

Participants were asked about their perspectives on strategies to tackle factors that impede participation in PA among individuals with T2DM. Most participants suggested that patient education targeting knowledge gaps and misconceptions holds great promise for promoting PA in this population.


*A lot of work needs to be done on patient education….. we need to empower them to make the right choices [in managing their condition]. I think, as nurses, we should find ways to educate them well.*
—N2

## 4. Discussion

In this study, we explored the opinions of HCPs regarding barriers and facilitators to PA participation in Ghanaian adults with T2DM. Overall, the findings demonstrate that both patient-related factors and healthcare system-related factors play a crucial role in shaping PA behaviour in this population. The barriers identified include inadequate accessibility to physical therapists and therapy centres, lack of knowledge and training among nurses and doctors on effective PA interventions for T2DM, time constraints during patient consultations, high cost associated with accessing physical therapy, patients’ reliance on doctors’ advice while disregarding PA guidance from other HCPs, and the perception among some patients that PA is not relevant for diabetes treatment. However, HCPs expressed belief in potential facilitators, such as integrating physical therapists and diabetes educators into diabetes care, providing structured exercise resources, improving curriculum planning to prioritize PA in health science education, and addressing knowledge gaps and misconceptions.

The research findings emphasise the challenge of inadequate access to physical therapists and therapy centres within Ghana’s healthcare system, hindering patients from meeting their PA guidelines. This problem creates a gap in the availability of professional guidance and support for individuals with a need for PA-related care. Given Ghana’s status as a low–middle-income country with a constrained health budget and structural challenges in its healthcare system [34], the lack of accessible physical therapy services compounds the existing healthcare limitations. Physical therapists are educated to have a comprehensive understanding of diabetes and its related conditions, capable of prescribing PA and addressing important factors that can impact an individual’s ability to engage in PA [35]. Recognizing the role of clinical exercise physiologists and establishing reimbursement mechanisms or specific programs, Western countries have successfully integrated clinical exercise physiology services into the broader continuum of care [36]. For instance, Australia has implemented a proactive health policy initiative, approving exercise physiologists’ role in primary and secondary exercise physiology services with a general practitioner referral, integrating them into the healthcare system seamlessly [37]. If Ghana achieves similar integration, combined with incorporating exercise physiology services into the National Health Insurance Scheme, these actions may potentially improve diabetes care standards.

There is evidence that access to exercise therapy is strongly related to increased PA in adults [38]. Access cost is also reported to affect the utilization of exercise therapy services in populations with low SE status [38]. The finding is consistent with studies in both low- and high-income countries which identified cost as a barrier to PA participation [39,40]. Therefore, developing financially viable PA programs that consider the socioeconomic status of individuals can maximize participation and address cost-related barriers to PA engagement.

The finding that patients with T2DM highly value doctors’ opinions is significant and reflects the trust and confidence they have in their primary healthcare providers. Similar to a study undertaken in India, 77% of patients with T2DM preferred to receive medical advice from their medical doctor instead of other HCPs [41]. While this trust in doctors is important, it is also essential to recognize the unique role that exercise physiologists play in the management of diabetes. Exercise physiologists possess specialized knowledge and expertise in prescribing exercise interventions, designing personalized physical activity programs, and addressing the specific needs and challenges faced by individuals with diabetes [35]. Their understanding of the physiological and metabolic responses to exercise enables them to provide targeted guidance and support for diabetes management [35]. Despite evidence that PA programs designed and implemented by exercise physiologists significantly improve exercise behaviour, HCPs in Ghana rarely refer clients to see exercise clinical physiologists [42].

To reconcile the trust that patients place in doctors with the involvement of exercise physiologists, a collaborative and multidisciplinary approach should be adopted. This can be achieved by developing collaborative guidelines or protocols that outline the appropriate indications for referral to clinical exercise physiologists [43], as well as implementing integrated care models [44] where doctors and clinical physiologists work within the same healthcare setting. Consistent with previous research findings [30,31], this study also revealed that doctors face time constraints, mainly due to a high volume of patients, which limits their ability to discuss PA during patient consultation. There is, therefore, the need for a collaborative effort, given that a collaborative team has a significant influence on behaviour change for patients with chronic diseases including T2DM [45]. There is evidence that stronger collaboration between multidisciplinary healthcare teams and clients to plan their PA and ongoing follow-up will result in positive patient outcomes [45].

A significant finding of this study is that HCPs lack knowledge regarding best practices for exercise in people with T2DM. Prior research has found that HCPs lack knowledge about PA guidelines and how to support patients to incorporate them into daily activities [46]. A systematic review conducted on primary care providers’ perspectives regarding exercise prescription in their practices identified major barriers such as lack of knowledge or training in exercise prescription [47]. Additionally, a study in Ghana shows that less than 20% of HCPs had confidence in undertaking PA assessment and counselling [42].

Findings from this current study underscore the critical need to prioritize education and training initiatives for HCPs to ensure they possess the necessary knowledge and skills to effectively integrate exercise into the management of T2DM. Some evidence suggests that continuous professional development for some HCP groups, especially nurses and doctors, improves their competence and confidence in exercise prescription [48]. Findings from this study also suggest that HCPs are optimistic that incorporating comprehensive and evidence-based PA education into medical and nursing curricula will equip them with the necessary knowledge and skills needed to deliver effective PA guidance and support to patients. The inadequacy of PA content in medical school curricula has been reported in other countries including Australia [49], the United States [50] and the United Kingdom [51]. The United Kingdom provides an average of 4.2 h of PA training throughout the entire four-year medical degree curriculum, while the United States offers approximately 8.1 h of PA training [52].

Attempts should be made to integrate PA training within medical and nursing school curricula. In Australia, for example, a study conducted among 19 Australian medical schools reported that at least about 87% of those schools taught the national aerobic guidelines, although the majority of the schools did not train students on the national strength training recommendation [49]. In Ghana, the current curricula of the University of Ghana Medical School and the School of Nursing lack the inclusion of PA content or training for their students [53]. Improving the curriculum to include PA as an essential component of health science education has the potential to effectively address the promotion and prescription of PA within clinical contexts.

Our study also found that misconceptions about PA among people with T2DM are a major barrier to PA participation. Some studies have reported inadequate knowledge about PA as a barrier to PA participation in people with T2DM [40,54]. These misconceptions may be influenced by social and cultural beliefs. For instance, HCPs explained that patients place more emphasis on traditional medicine, including pharmacotherapy, resulting in the neglect of other effective strategies, such as PA. Arias et al. [55] found that herbal and spiritual treatments are the first treatment options for individuals in Ghana. Findings from other studies in Ghana also suggested that some patients consider PA as not important in the management of T2DM [8]. The findings from this study underscore the urgent need for educational interventions to address these misconceptions and improve knowledge among patients. By targeting these misconceptions and providing accurate information on the benefits of PA in T2DM management, educational interventions can help overcome the barriers associated with misconceptions and promote the adoption of PA as an integral part of T2DM management. Such interventions should consider the cultural and social context to effectively address the specific misconceptions prevalent in the population.

Our finding underscores the importance of implementing strategic policies and initiatives to address the structural challenges in the healthcare system. Particularly, allocating resources to expand the availability of therapy centres and training more physical therapists to bridge the accessibility gap and ensure individuals receive the necessary support for their PA requirements. Ghana should prioritize the integration of these services within the broader healthcare system. Moreover, it is strongly recommended to enhance collaboration among multidisciplinary healthcare teams, establish an efficient referral system for PA services, and improve the affordability and accessibility of these services. Further, consideration of alternative forms of PA, including home-based exercise, may be worthy of consideration for patients with T2DM. In support of this, there is some evidence that home-based programs may be as effective as supervised facility-based programs [56]. Preliminary evidence from a feasibility randomized controlled trial with Ghanaian adults with T2DM indicates that a low-cost home-based PA program is feasible, safe, and has high acceptability in that population [24]. Similarly, the Diabetes em Movimento, a low-cost, community-based group PA program implemented in Portugal for people with T2DM, has shown promising results [57]. Future research should focus on developing culturally tailored interventions that address patient and socio-structural barriers and facilitators identified in this study.

Interpretation of the findings of this study should be considered within the limitations of this study. We examined HCP opinions about barriers and facilitators to PA among people with T2DM in Ghana. Consumers’ opinions are needed, and the subject of an additional study will be published separately. Our findings do not transcend all social contexts in Ghana because the various regions have different cultures. Participants were recruited from the southern and middle belts of Ghana; hence, the findings may be relevant to patients living in those regions of Ghana.

## 5. Conclusions

This study provides valuable insights into the barriers and facilitators of PA participation among Ghanaian adults with T2DM, from the perspective of HCPs. Inadequate accessibility to physical therapists and therapy centres, insufficient knowledge and training among healthcare practitioners, time constraints during patient consultations, and the cost of PA services hinder the provision of PA interventions. Patient factors, such as reliance on doctors and misconceptions about the relevance of PA, further contribute to the challenges. However, integrating physical therapists and diabetes educators, providing structured exercise resources, improving health science education curricula, and addressing knowledge gaps and misconceptions were identified as potential facilitators. In the future, this information may guide the design of a PA program for Ghanaian adults with T2DM.

## Figures and Tables

**Table 1 sports-11-00123-t001:** Characteristics of the sample.

Characteristics	N [%] [n = 13]
Sex	
Male	6 [46.2%]
Female	7 [53.8%]
Age, years	
18–35	8 [61.5%]
36–55	3 [23.1%]
Above 55	2 [15.4%]
Profession	
Exercise physiologist	3 [23.1%]
Nurse	3 [23.1%]
Physician	4 [30.7%]
Physiotherapist	3 [23.1%]
Working experience, years	
3–4	1 [7.7%]
5–8	8 [61.5%]
>8	4 [30.8%]
Affiliated institution	
Clinical setting	7 [53.8%]
Research institution	3 [23.1%]
Both clinical and research	3 [23.1%]

N = total number of participants.

**Table 2 sports-11-00123-t002:** Major themes and sub-themes.

Main Themes	Sub-Themes	
	Barriers	Facilitators
Health system-related factors	Time constraints to promote PA during patient consultation Inadequate access to exercise therapists, programs and therapy centres affects PA participation Lack of clear-cut PA guidelines from Ghana Health Service for patients with diabetes Cost of accessing physical therapy presents a significant challenge in achieving PA goals	Promoting the role of physical therapists and diabetes educators in diabetes care Improving curriculum to include PA as an essential component of healthcare education Providing readily available structured exercise resource
Healthcare practitioner factors	Lack of evidence-based knowledge on PA guidelines HCPs over-emphasise medication and diet in diabetes management	Continuous professional development is needed for nurses and doctors
Patient factors	Disregard of PA advice from some members of the healthcare team Perception that PA is not a treatment option for diabetes	Addressing knowledge gaps and PA misconceptions

PA = physical activity, HCPs = healthcare professionals.

## Data Availability

The data presented in this study are available on request from the corresponding author.

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
