# Peer review of "Expert Opinions about Barriers and Facilitators to Physical Activity Participation in Ghanaian Adults with Type 2 Diabetes: A Qualitative Descriptive Study"

_sports, 2023, doi:10.3390/sports11070123_

Round 1

Reviewer 1 Report

Dear authors, congratulations on your work. It's interesting and brings great contribute to the field. With these results, a new way should be taken to promote more physical activity behaviour in this population. However, I intend that there are some points that should be solved before publication of this work.

Abstract

Line 15 – T2DM which men? Type 2 diabetes M (mellitus?). Please update like introduction section.

Line 20 – Three main themes of?

Introduction

This section is well-designed, that identified the real importance of this theme. Would it be also important to realize the outcome on the quality of life of T2DM subjects of PA. You identified that PA improves resistance to insulin… i.e. Specific things. But what is promoted? More quality of life...well-being, more health? Some data related to this will improve this section. It is also significant to talk about the economic concerns associated with T2DM, and the improves that could be made if more T2DM subjects were more active.

Methods

Results

Table 1. Add info about female N

Discussion / Conclusion

Interesting discussion was made. To create more impact of this section, you could aboard about some programs that promote PA behaviour in this population, as "Diabetes em Movimento" in Portugal" It could be an example to implement in African context.

Recommendations for future research should be added.

Author Response

Thank you for your feedback. We appreciate your comments.

Reviewer 2 Report

Without a doubt, physical activity is a good treatment for type II diabetes. In addition, in African countries there is an alarming increase in this pandemic, therefore this type of research is highly relevant.

I am going to make a series of suggestions to the authors, which in my opinion, would improve the paper.

Line 68-69 the sentence “In the 68 future, this information may inform the design of a physical activity program for this population. ” should be moved to the end of the conclusion, it is certainly an interesting recommendation.

Lines 84-85 consider that the information on the characteristics of the country should be included in the introduction, it does not make much sense for it to appear in the method.

Lines 139 the information on the duration of the interviews already appears in line 115, the scientific literature must obey the principle of economy, therefore, this information must appear in one place.

In order to make reading easier, I recommend authors start the discussion by remembering the objective.

The conclusions should be more specific, indicate what the "barriers and facilitators" are.

Reviewer 3 Report

The aim of this study was to explore opinions of healthcare professionals regarding barriers and facilitators to physical activity participation among people with Type 2 diabetes mellitus in Ghana.

It is an interesting study that addresses the different levels of barriers to physical activity for this type of patient. Despite being a small sample, the study is well designed, but some aspects of its wording need to be improved.

Here are my contributions:

- It is necessary to complete the introduction by adding information on barriers to physical activity in the general (non-diseased) population of Ghana or other population groups with other types of diseases.

  • Line 77 shows an inappropriate reference format.

  • What is the socioeconomic status of the patients treated by the professionals interviewed for the study? It is necessary to include information on the locations of the centers where the surveyed professionals work. 

  • In Table 1, why is there only one sex?

  • It is necessary to explain below the tables, what each abbreviation included in them refers to.

  • Line 263, the paragraph is not 3.1, is it?

  • Line 321-334. Such conclusions or practical applications should come at the end of the discussion. 

Round 2

Reviewer 1 Report

Thank you for addressing my comments.

Congratulations.

Reviewer 3 Report

Congratulations to the authors for the improvements made to the manuscript.